# What Strategies Do Healthcare Providers Use to Promote Adolescents’ Vaping Cessation? A Scoping Review

**DOI:** 10.3390/ijerph22060839

**Published:** 2025-05-27

**Authors:** Naima Nimmi, Bindu Joseph, Habib Bhurawala, Smita Shah, Anita Munoz, Muhammad Aziz Rahman

**Affiliations:** 1Institute of Health and Wellbeing, Federation University Australia, Berwick, VIC 3806, Australia; b.joseph@federation.edu.au (B.J.); ma.rahman@federation.edu.au (M.A.R.); 2Paediatric Department, Nepean Hospital, Nepean Blue Mountains Local Health District, Penrith, NSW 2750, Australia; habib.bhurawala@health.nsw.gov.au; 3Paediatrics, Nepean Clinical School, Faculty of Medicine and Health, The University of Sydney, Sydney, NSW 2050, Australia; 4Paediatrics, School of Medicine, The University of Notre Dame Australia, Sydney, NSW 2007, Australia; 5The Woolcock Institute of Medical Research, Macquarie University, Sydney, NSW 2113, Australia; smita.shah@health.nsw.gov.au; 6Prevention Education and Research Unit, Western Sydney Local Health District, North Parramatta, NSW 2151, Australia; 7School of Public Health, Faculty of Medicine and Health, The University of Sydney, Sydney, NSW 2050, Australia; 8Healthcare on Collins, Melbourne, VIC 3000, Australia; anitamunoz78@gmail.com; 9Faculty of Public Health, Universitas Airlangga, Surabaya 60115, Indonesia

**Keywords:** healthcare providers, e-cigarettes, adolescents, practices

## Abstract

Background: Healthcare providers (HCPs) can play a pivotal role in providing vaping cessation support to adolescents. They can screen adolescents for vaping, offer interventions to quit, and educate adolescents about the dangers of vaping, including nicotine addiction. This review aims to assess the knowledge, attitudes, and practices of HCPs regarding vaping cessation promotion to adolescents. Methods: We conducted a scoping review following the Arksey and O’Malley framework and searched five databases, including MEDLINE, PubMed, PsycINFO, CINAHL, and Scopus, to identify relevant articles. We created a comprehensive search strategy using keywords relevant to healthcare providers, adolescents, practices, and vaping. Results: A total of 1387 articles were identified from the initial search, and 14 were included. There was considerable variation regarding the knowledge, attitudes, and practices of HCPs in supporting adolescents. Although 50–92% of HCPs reported some knowledge about vaping, significant gaps persisted, especially regarding health effects and cessation strategies. Many HCPs lacked confidence in discussing vaping with adolescent patients. While most of the HCPs, 86%, screened for cigarette smoking, only 14% routinely screened adolescents for vaping. They expressed concerns about vaping as a gateway to tobacco use and believed that it was less harmful than cigarettes. HCPs encountered barriers with a lack of time, knowledge, and screening tools in supporting adolescents to quit vaping, particularly related to counselling, treatment recommendations, and referral procedures. Conclusions: There is an urgent need for enhanced understanding, evidence-based guidelines, and clinical tools for HCPs to address the current vaping epidemic among adolescents.

## 1. Introduction

Vaping or electronic cigarette (e-cigarette) use has rapidly increased worldwide and is recognised as a global public health concern [1]. E-cigarettes are a distinct group of battery-operated handheld devices that vaporise liquid solutions when heated to a certain temperature. Users inhale using a mouthpiece [2]. Although the composition of aerosol-producing e-cigarettes or vaping products varies, they typically contain propylene glycol, vegetable glycerine, a variety of flavourings, and nicotine [3].

The prevalence of cigarette smoking among youth has continued to decline while vaping has been increasing, particularly over the last decade [4]. Studies from various countries indicate that vaping is increasing among youth who never smoked [3]. The global prevalence of lifetime vaping among adolescents is 19.9%, and the current rate of use is 8.8% [5]. Increased rates of vaping have also been reported among adolescents in Greece, Romania, the UK, Canada, New Zealand, and Australia [6]. From 2011 to 2020, the prevalence of vaping amongst US high school students increased from 1.5% to 19.6% [7]. According to the 2022–23 National Drug Strategy Household Survey, the proportion of Australians aged 14 and over who had ever used an e-cigarette rose to 19.8%, continuing the upward trend from 4.9% in 2016 and 6.9% in 2019 among non-smokers [3,8]. Findings from the Pacific region also indicate that the number of people who vape is increasing; however, clear trends among adolescents are still yet to be identified. This rising prevalence of vaping among adolescents highlights the urgent need for healthcare professionals, especially those in primary and paediatric care, to play a proactive role in screening, counselling, and guiding adolescents on vaping-related harms. Despite the scale of the problem, HCPs’ engagement in vaping cessation remains inconsistent and poorly defined in current practices.

Manufacturers continue to promote new devices and attractive packaging for vaping products, which often target teenagers and young adolescents [9]. More than 30,000 vaping brands have been identified in the EU region, and approximately 16,000 unique flavours are now available worldwide, many of which are very appealing to teenagers and young adolescents [7]. However, these devices usually contain nicotine, which is addictive, raising concerns about vaping and nicotine addiction in teenagers and young adolescents [3,10]. There is also documented evidence of harms related to inhalation of toxic chemicals in vaping liquid and the potential for nicotine overdose with vaping [11,12]. Nicotine consumption also increases the risk of delayed brain development and potentially affects learning ability among young adolescents [2,13]. Therefore, minimising nicotine exposure from any form of tobacco product in adolescents and youth is crucial.

HCPs have the potential to intervene in reducing vaping amongst adolescents, as they do with other risky behaviours in this age group [14,15]. Given the growing inclination towards vaping among adolescents, HCPs must understand the negative health impacts of vaping as well as recognising that e-cigarettes or vapes are not solely a smoking cessation device designed for adult use; their knowledge and practices should not be restricted to solely this application [2]. HCPs are critical frontline professionals in adolescent healthcare. As the first point of contact in primary care settings, they are well positioned to deliver preventive interventions, assess risk behaviours, and provide early counselling for nicotine dependence, including vaping. Evidence shows that HCP-delivered counselling increases quit rates and reduces relapse in adolescent smokers [16]. However, their role in vaping cessation remains underdeveloped. Although the included studies in this review did not apply specific theoretical models, behavioural constructs such as HCPs’ attitudes, confidence, and perceptions of social norms may influence clinical engagement. These elements are reflected in behavioural science frameworks like the Theory of Planned Behaviour (TPB) and Social Cognitive Theory (SCT), which may help explain why some HCPs engage in vaping cessation and others do not, despite similar exposure. Future research is needed to explore these theoretical perspectives empirically.

Knowledge, attitudes, and practices around vaping, particularly with adolescents, are limited. Conflicting evidence about the use of e-cigarettes as a cessation tool and evolving regulatory frameworks further contribute to uncertainty in clinical practice. These gaps, along with a historical emphasis on cigarette smoking over vaping, highlight the urgent need to examine HCPs’ preparedness to support adolescent vaping cessation and identify areas for targeted training, clearer clinical pathways, and improved support tools. Although prior studies have addressed youth-focused messaging, public health interventions, and general cessation strategies, the role of HCPs remains underexplored. Prior studies on prevention strategies [17], health messaging [18], and public health policies [19,20] did not assess how HCPs counsel and intervene in adolescent vaping. Additionally, school-based programs and public health initiatives have predominantly been explored [21,22]. While some studies have analysed HCPs’ beliefs about e-cigarettes [23,24], they do not provide insights into variations across different specialties of HCPs.

Our study aims to identify key barriers that HCPs face in counselling strategies as such challenges have not been analysed systematically in previous studies. In contrast to prior research focusing solely on physicians, our review compares different types of HCPs. Therefore, our aim was to conduct a scoping review to assess the knowledge, attitudes, and practices of HCPs in providing vaping cessation support to adolescents. By identifying gaps in HCPs’ knowledge, screening methods, and counselling strategies, our study focuses on the urgent need for practice changes. Integrating vaping cessation into routine practice and offering structured HCP training will address a critical gap in the literature and have direct implications for improving adolescent health outcomes.

To capture a comprehensive range of evidence-based studies while adhering to the PCC (Population, Concept, and Context) framework, we formulated the following research question for this scoping review (Table 1):

What strategies do healthcare providers use to promote adolescents’ vaping cessation?

## 2. Materials and Methods

This scoping review followed the Arksey and O’Malley framework from the Joanna Briggs Institute Reviewer’s Manual [25]. This review also followed the guidelines for the Preferred Reporting Items for Systematic Reviews and Meta-Analyses extension for Scoping Reviews (PRISMA-ScR) [26]. A systematic search strategy was developed in consultation with the university librarian, using a comprehensive set of keywords to identify peer-reviewed articles across multiple databases. The scoping review methodology was applied, which allowed the mapping of the main concepts and provided an overview of the evidence available for the research area. Arksey and O’Malley’s scoping review methodology consists of six components: (i) identifying the research question, (ii) identifying relevant studies, (iii) selecting studies, (iv) charting the data, (v) collating, summarising, and reporting the results, (vi) consulting with key stakeholders (optional) [25]. We did not include consultation as part of the current scoping review, as the review focuses on using terms in the literature rather than seeking broader insights into timeframes and delays. Additionally, this scoping review did not include a formal stakeholder consultation, a recommended but optional stage in the Arksey and O’Malley framework. The primary aim was to map peer-reviewed literature rather than co-develop practice recommendations. However, our investigator team included a general practitioner and a paediatrician, who were closely involved in framing the research questions, interpreting findings, and refining implications. Their involvement provided relevant professional perspectives from frontline healthcare practice. The absence of broader consultation with stakeholders such as adolescents, parents, educators, and policymakers may have limited our understanding of contextual implementation challenges. Future studies are encouraged to incorporate formal stakeholder engagement to ensure that the development of interventions and policies is responsive to diverse contextual needs.

### 2.1. Databases and Search Strategy

We searched MEDLINE, PubMed, PsycINFO, CINAHL, and Scopus to identify relevant studies up to November 2024. The keywords used in the search strategy were based on relevant keywords and medical subject headings (MeSH) terms. Search terms were “health care providers” OR “professionals” OR “doctor” OR “nurse” OR “physician” OR “paediatrician” OR “primary health care provider” AND “e-cigarettes” OR “electronic cigarettes” OR “vapour cigarettes” OR “vapes” OR “electronic nicotine delivery device” OR “Juul” OR “vaping” AND “adolescents” OR “teenagers” OR “young adults” OR “teen” OR “youth” OR “high school students” OR “middle school students” AND “practices” OR “strategies” OR “approaches” OR “knowledge” OR “attitude” OR “perception”. Full search strings are included in Appendix A. We also searched the reference lists of key publications to retrieve additional articles. Searches from databases were documented after completing the initial searches. The references were imported into the bibliographic software EndNote™ X10 to identify duplications, and further screenings were completed in Covidence [27].

### 2.2. Inclusion and Exclusion Criteria

We included original peer-reviewed articles published in English, irrespective of their design (quantitative or qualitative) and methodology, and studies focused on HCPs who supported adolescents for vaping. “Support” was defined as any clinical action, including screening for vaping use, counselling, provision of educational information, behavioural interventions, or referral to cessation services. “Adolescents” were defined as individuals aged approximately 10–19 years, consistent with the World Health Organization’s definition. Studies included if the target population fell entirely or predominantly within this age group. Commentaries, reports, and reviews were excluded from the analysis. The population of the HCPs included practitioners from all healthcare disciplines, including medicine (general practitioners, paediatricians), nursing, surgery, dentistry, pharmacy, psychology, or allied health professions that have provided any vaping cessation support to their adolescent patients or clients. As vaping was predominantly prescribed for adults, studies specifically investigating the use of e-cigarettes as a smoking cessation aid were excluded. Therefore, this review paper focuses not on the efficacy of vaping prescribed for smoking cessation but rather on other aspects related to HCPs. Additionally, studies were excluded under the category of “wrong outcome” if they did not address HCPs’ views, practices, knowledge, or support strategies related to adolescent vaping cessation or if the outcome measured was irrelevant, incomplete, or incompatible with the aims of the review. Studies that primarily examined the smoking habits of HCPs themselves were also excluded.

### 2.3. Selection of Articles

One independent reviewer (NN) conducted the title and abstract screening. Three reviewers (NN, AR, and BJ) independently conducted the full-text review of the eligible articles. Any disagreement regarding the selection of publications was resolved through discussion amongst the co-authors.

### 2.4. Data Extraction

The first author (NN) carried out the data extraction based on the research question and reviewed by other co-authors. Data on study characteristics and findings were extracted into an Excel spreadsheet. Extracted data included information on study design, setting, location, year of publication, methods, population, HCP types, age of the adolescents, supports provided, outcomes related to knowledge, attitudes, and practices of the HCPs, and limitations. The extracted data were summarised into Table 2 and Appendix A and reported narratively.

### 2.5. Quality Assessment

Fourteen peer-reviewed articles were selected in the final search to maintain high scientific standards in the content. Three authors independently reviewed all full-text articles, including them in the final selection based on predefined criteria. Any disagreements or doubts were resolved through discussion amongst the co-authors. The quality of the selected papers was assessed using the Critical Appraisal Skills Programme (CASP, 2023) [28], a tool that applies a structured set of comprehensive questions tailored to specific research methods. The methodological quality of the included studies was assessed using the Critical Appraisal Skills Programme (CASP, 2023) checklist. Three reviewers evaluated each study independently (NN, MAR, BJ) across ten CASP criteria, including study design, recruitment, ethics, data collection, data analysis, findings, and research value.

Based on the assessment, studies were assigned an overall quality rating [28]: A = nil or few flaws, credibility and transferability high; B = some flaws, unlikely to affect credibility significantly; C = flaws that may affect credibility; D = major flaws likely affecting credibility (none found in this review). In cases where initial reviewer assessments differed, disagreements were resolved through consensus discussion among all three reviewers. Full appraisal outcomes for each included study are detailed in Appendix A. The evaluation process was transparent, with the focus on key areas such as clear aims, methodology, design, recruitment, data collection, ethical considerations, data analysis, findings, and research value.

## 3. Results

The initial search yielded 1387 articles. After removing 339 duplicates, the titles and abstracts of 1048 articles were screened to identify articles for full-text review. After the exclusion, the remaining 84 articles were eligible for full-text review, and 14 studies were included in this scoping review (Figure 1).

### 3.1. Study Characteristics

This review included ten cross-sectional studies, two qualitative studies, one mixed-method study, and one retrospective cohort study (Table 2). The most commonly included HCPs were family medicine physicians, paediatricians, nurse practitioners, and physician assistants and other allied health professionals. The ages of the adolescents receiving support in the included studies were inconsistent, and some studies did not specify the age of adolescents. The summary characteristics of the included studies are outlined in Table 2. The views of the HCPs regarding support for adolescents for vaping varied widely with different outcome definitions (e.g., knowledge, counselling, clinical skills) in the selected studies. Some studies directly assessed beliefs, knowledge, and attitudes related to queries about vaping, using Likert scales from one to five, with responses varying from “strongly disagree” to “strongly agree” [4,11,23,29]. Another two studies assessed the comfort level with different behaviour queries, including “talking to a patient about vaping”, with responses ranging from one to four: “least comfortable” to “most comfortable [4,23]. Responses from other studies were a combination of open- and close-ended questions, which were coded as negative, neutral, or positive in attitude towards vaping [11,30].

Most of the included qualitative studies used interview guides varying in content, but all contained questions related to three primary domains: investigating (a) general knowledge, (b) perceptions, attitudes, and clinical experience, and (c) practices concerning adolescent vaping amongst HCPs. To investigate each domain, the HCPs who participated in those studies were asked a generic question at first (“In your practice, how have you dealt with e-cigarettes?”), followed by additional questions to identify specific information (for example, “Have you ever been questioned by a patient about e-cigarettes? If so, can you describe the conversation and the advice you provided?”) [31].

### 3.2. Knowledge of HCPs About Vaping by Adolescents

Although most of the included studies reported perceived knowledge of vaping among HCPs, the knowledge assessment tool was highly heterogeneous. Many HCPs, ranging from 50% to 92%, had a certain level of knowledge regarding the harm caused compared to traditional cigarettes [23,29,30,31,32,33,34,35]. Several studies indicated that HCPs had limited knowledge of the potential health risks associated with vaping [23,31], whilst HCPs from another study reported having knowledge about addictiveness (51%), nicotine content (81%), and availability of flavoured e-cigarettes (43%) [31]. A recent study found that HCPs perceived a significant difference (*p* < 0.001) in addictiveness and harmfulness between cigarettes and vaping [36]. HCPs were also interested in learning about the health effects of passive and active vaping [11,24,31]. However, a considerable knowledge gap exists among HCPs in helping patients with vaping cessation compared to helping patients with quitting smoking [11]. In addition, an overall knowledge gap was also found regarding the chemical content of e-cigarettes, symptoms of e-cigarette or vaping-related lung injury (EVALI), and types of e-cigarettes [34].

**Table 2 ijerph-22-00839-t002:** Summary characteristics of the included studies.

First Author (Year)	Location; Sample Size	Sampling Technique and Data Collection Method	Types of HCPs	Age of the Adolescents	Support Provided for Vaping E-Cigarette Use by the HCPs
**Cross-sectional studies**					
Rajiv Singh et al. (2024) [37]	AUS; 53	Convenience sample, online surveys through publicly available phone numbers and email addresses	General practitioner and general practitioner trainee	12–17	Providing care for children and adolescents
Sundstrom et al. (2023) [35]	USA; 257	Convenience sample, online surveys via social media	Clinical dental hygienists	Did not specify	Asked patients about e-cigarette use
Chin et al. (2022) [29]	USA; 64	Purposive sampling, via email to the Paediatric Endocrine Society	Attending, fellow, nurse practitioner, physician assistant	Providers reported screening at 10–16	Screened patients about e-cigarette use
Gorukanti et al. (2022) [36]	USA; 771	Online survey	General internal medicine physicians, paediatricians, and family medicine physicians	10–17	Provided screening for e-cigarette use
Metcalf et al. (2022) [11]	USA; 31	Purposive sampling, direct email to the providers	Physicians, nurses, social workers, psychologists, epidemiologist	Did not specify	Talking to the parents of adolescents about vaping prevention or helping them to quit
Cano Rodriguez et al. (2021) [9]	USA; 11	Weekly chart reviews of all adolescent visits with a primary care provider	Adolescent medicine physicians, nurse practitioners, nurses, patient care associates, social workers, general paediatricians	Did not specify	Provided screening on vaping to adolescents
McGee et al. (2021) [34]	USA; 169	Self-administered, anonymous survey	Family medicine physician, nurse practitioner, registered nurse, licenced vocational nurse, medical assistant technician patient care	Did not specify	Provided care for adolescents who vaped
Simoneau et al. (2021) [32]	USA; 95	Convenience samples, surveys, and focus group discussion	Paediatrician	11.6 ± 2.1	Provided counselling to adolescents
Pepper et al. (2015) [24]	USA; 776	Online survey	Paediatrician, family medicine physician	11–12	Routinely screened by the physicians
Pepper et al. (2014) [23]	USA; 561	Online survey	Family medicine physicians, paediatricians, nurses practitioners	11–17	Provided preventive care to adolescents
**Retrospective cohort study**					
Oliver et al. (2022) [4]	USA; 447	Purposive sampling promotional email to the provider	Advanced nurse practitioner, certified prevention specialist, clinical psychologist, community health worker, licenced clinical addiction counsellor, primary care physician, psychiatrist, social worker	Did not specify	Provided treatment to adolescents who vaped
**Qualitative studies**					
Peterson et al. (2018) [33]	USA; 25	Purposive sampling; Skype, face to face	Family medicine physician, paediatricians, nurse practitioner, physician assistant	13–19	Provided counselling to adolescents
Brown-Johnson et al. (2016) [30]	USA; 72,000	Data was publicly available from Health Tap *	Family medicine physician, mental health practitioner, dentistry	Answer provided to the adolescent	Provided counselling to adolescents
Gorzkowski et al. (2016) [31]	USA; 37	Focus group discussion	Paediatrician	Did not specify	Provided care to the adolescents

* an online patient–provider digital health services with a repository of anonymous patient questions and public answers from approximately 72,000.

#### Knowledge of Vaping: Differences by Healthcare Provider

Family medicine physicians and GPs demonstrated awareness of nicotine harm but lacked knowledge of chemical contents and vape device types [4,22,27,30,31,32,33]. Paediatricians reported a limited understanding of adolescent-specific risks and were less familiar with EVALI symptoms or vape constituents [4,10]. Nurse practitioners and other allied health professionals indicated moderate knowledge levels and a strong interest in further education [10,22].

### 3.3. Beliefs of HCPs About Vaping by Adolescents

Overall, the included studies found that HCPs commonly believed that adolescent vaping might be a “gateway” to smoking [4,23,24,29,33,36,37], but there was conflicting evidence about the safety of vaping. Whilst most of them believed that e-cigarettes were safer than combustible tobacco [4,23,31], some argued that they were equally or more dangerous than cigarettes [32]. Around 80% of HCPs believed that nicotine-containing vaping was a serious problem for youth [29]. Providers who considered vaping as safer were less likely to consider discussing it with patients [23,24], and some did not believe that discussing tobacco with adolescent patients had the potential to change their behaviour [29]. Only 37% of HCPs received support from senior HCP supervisors in inquiring about adolescent vaping during their appointments [35].

### 3.4. Practices of HCPs in Vaping Cessation Support for Adolescents

Although routine screening for vaping was largely missing among all fourteen studies, HCPs reported supporting their adolescent patients through infrequent screening [24,29,36], counselling [11,30,32,33,34,36], and providing treatment [4,9,23,31]. The most commonly used management approaches to adolescents were education about the risks of vaping [24,29,36], brief motivational interviewing [4,33], and discussion of harms when assisting vaping cessation [29,32]. However, routine screening for vaping use was less common compared to screening for cigarette smoking (86% vs. 14%; *p* < 0.001) [24,31], and routine counselling and assistance in quitting were more common for cigarette smoking than for vaping (79% vs. 18%; *p* < 0.01) [29]. In a recent study, HCPs screened 50% of adolescents for vaping, whilst they screened 100% of adolescent patients for traditional cigarette use; they also provided counselling to 90% of traditional cigarette users compared to 20% of the patients who vaped, with the difference in screening and counselling approaches being highly significant (*p* < 0.0001) [36]. In another study, HCPs discussed vaping with 34% of their adolescent patients, while discussions about conventional cigarettes occurred with 60% of patients [37]. HCPs reported various barriers to counselling, such as lack of time, knowledge, screening tools, and competing priorities. Some providers also reported not knowing where to refer and how to treat adolescents for vaping [29,31,34]. Additionally, while around 63% of providers felt they needed more time to ask about vaping, only 33% agreed they needed more resources to do so [35]. A study found that older HCPs felt more comfortable discussing vaping with their adolescent patients, and family medicine physicians were more proactive in discouraging the use of vaping compared to paediatricians [23].

#### Vaping Cessation Practices Stratified by Geography

Most of the studies on HCPs’ vaping cessation practices were from the USA, showing variable practices with low rates of routine screening and counselling. For example, only 34% discussed vaping with adolescents vs. 60% for cigarettes [20]. Similarly, only a pilot study conducted in Australia showed that GPs screened inconsistently and rarely offered targeted vaping counselling [37].

### 3.5. Training and Other Related Findings of HCPs

Four studies reported the need for further training to improve clinical practice, with all HCPs agreeing on the need for upskilling to better treat vaping patients [11,23,29,31]. HCPs also expressed interest in training in communication skills, about the prevention of vaping, and in helping adolescents to quit [11]. Two studies reported that training significantly improved the screening and comfort levels in discussing vaping and counselling adolescent patients, with comfort rising from 0% from the initial session to 100% by the sixth session (*p* < 0.05) [4,9].

Limited data suggested that the primary sources of information on vaping for HCPs were patients, news stories, and advertisements rather than professional sources [29]. This statement highlights a potential gap in the availability or accessibility of reliable, evidence-based professional resources on vaping for HCPs, which is crucial for effective patient care.

Overall, the approach of HCPs to adolescents when screening for and counselling about vaping was inconsistent across studies and influenced by their awareness, knowledge, beliefs, and attitudes towards vaping among adolescents. HCPs often advised adolescents without themselves having sufficient knowledge about vaping and did not provide suitable screening, counselling, and cessation support.

## 4. Discussion

This is the first scoping review to synthesise the views of HCPs, including family medicine physicians, paediatricians, nurse practitioners, and physician assistants, regarding support for adolescents who vape. The review found that there was limited knowledge amongst HCPs about vaping, and most of them wanted to learn more to improve their counselling and treatment skills. Although HCPs had mixed beliefs on vaping, most of them believed that vaping could be a potential gateway to smoking in the future while also considering vaping safer than combustible cigarettes. Our review also found that routine screening for vaping was less frequent than for combustible cigarettes. The studies also found consistent evidence of barriers for HCPs in counselling, treating, and referring their adolescent patients, primarily due to inadequate training.

Our findings suggest that the majority of HCPs had limited knowledge about vaping despite having some familiarity with them, which aligns with previous studies [32,38]. Our review found that HCPs lacked awareness of nicotine content, addictiveness, and flavours associated with vaping [32]. Additionally, HCPs, including paediatricians, acknowledged their limited understanding of the potential health consequences of vaping [24,31,33], which is consistent with findings from another study [38]. A previous systematic review highlighted that HCPs received information from unreliable sources including news stories and advertisements, rather than scientific sources [39]. Due to a lack of knowledge of the harm of vaping and unfamiliarity with adolescent motivations for vaping, HCPs lacked confidence in discussing the issue with adolescents [33]. This is concerning because unreliable information can be passed on to patients, highlighting the need for unbiased, professional education for providers on the current state of vaping.

The HCPs held mixed beliefs towards vaping among adolescents. A total of four studies reported that HCPs believed vaping was safer than smoking combustible cigarettes [4,23,30]. Likewise, another major finding regarding the beliefs of HCPs, including paediatricians in half of the studies, was that vaping by adolescents might act as a “gateway” to smoking [4,23,24,29,33,36]. Both statements are consistent with prior findings from a systematic review [39]. Another study found that vaping increased the risk of ever using combustible cigarettes [3]. HCPs who believed that vaping was safer than other tobacco products were less likely to feel the importance of discussing vaping with patients [23]. On the other hand, HCPs also believed that nicotine-containing vaping devices were a significant problem for youth and reported that vaping was equally or even more harmful than cigarettes [23,32]. Another study demonstrated similar concerns held by HCPs in a systematic review [3]. One study reported that HCPs believed that quitting vaping was easier than quitting smoking [11].

Adolescent patient–provider interactions about vaping were reported in all thirteen studies. A diverse group of HCPs in these studies reported screening, counselling, and treating adolescent patients. However, almost half of the studies in our review found that few HCPs routinely screened and counselled their adolescent patients about vaping. In contrast, there were higher rates of screening and counselling services available for tobacco smoking [24,29,31,32,36]. This finding is consistent with the previous findings from a systematic review [39]. This gap in practice appears to be influenced by low self-efficacy, limited knowledge, and structural barriers, such as time constraints and lack of system-level support. While the studies reviewed did not apply behavioural theories explicitly, our findings align with constructs from the Theory of Planned Behaviour (TPB) and Social Cognitive Theory (SCT). TPB suggests that attitudes, perceived norms, and behavioural control shape intentions to act, while SCT focuses on the influence of self-efficacy and the clinical environment. These frameworks provide a useful lens to understand the variability in HCPs’ engagement in vaping cessation. However, further empirical research is required to explore these theoretical models in this context.

Moreover, preventive practices like discussing harms were also not practised routinely. Three studies reported barriers to counselling and referring their adolescent patients [29,31,34]. Similar barriers to referring and treating adolescent patients were reported in another study [34]. Most studies reported they had not received formal training, highlighting a need for such training. Among fourteen studies, two exclusively focused on training outcomes, and both observed a range of improvements following training and intervention [4,9]. Again, it was highlighted that short, brief training interventions can improve HCPs’ engagement in counselling [40]. One randomised trial found a statistically significant increase in the assessment of adolescent vaping for providers who received one-hour training in tobacco use prevention and cessation counselling [9].

However, current training programs often fail to address the unique challenges of adolescent vaping. Most are adapted from adult smoking cessation models and lack adolescent-specific competencies such as age-appropriate risk communication, understanding of psychosocial drivers of vaping, and use of non-judgmental counselling techniques. Moreover, training rarely covers practical elements like referral systems, brief interventions, or pharmacotherapy options for youth [4,39]. Structural barriers further limit the implementation of practical training in healthcare systems. These include limited time in clinical schedules and insufficient integration of vaping cessation into continuing professional development [39,40]. Additionally, the absence of national clinical guidelines on adolescent vaping undermines the standardisation and scaling of such training.

Regarding policy implications, this review supports the urgent development of national clinical guidelines for adolescent vaping cessation. Such guidelines for HCPs should include structured screening tools, brief intervention models, and age-appropriate counselling strategies. Furthermore, the identified training gaps emphasise a need for vaping-specific competencies within medical and nursing curricula. Public health campaigns should also align with clinical practice, using consistent messaging across school-based programs, general practice, and community health services to reinforce prevention and early intervention. These actions would support clinical decision-making and strengthen adolescent health policy by ensuring HCPs are well equipped to respond to an evolving public health issue.

### 4.1. Strength and Limitations

This study used a broad systematic search strategy in electronic databases, including diverse study designs, to review evidence in the research area. It highlights the diversity among HCPs from various backgrounds in addressing adolescent vaping cessation and identifies specific gaps in their knowledge, attitudes, and screening practices. The main limitations of this study include the predominance of studies from the United States, with a lack of data from low- and middle-income countries. One of the key limitations of this review was the substantial heterogeneity in measurement tools, outcome definitions, and conceptual framing across the included studies. The tools used to assess knowledge, attitudes, and practices varied widely from structured Likert-scale surveys to open-ended interviews, making it difficult to compare results across studies directly. Definitions of key outcomes such as “screening”, “support”, or “counselling” were also varied across the studies. Furthermore, adolescent age ranges were not uniformly defined, and several studies did not specify the age ranges. This variation compromises the ability to synthesise findings reliably and limits the generalisability of the observed patterns. Future research in this area would benefit from standardised definitions and validated tools to enable more robust comparisons across healthcare provider groups and settings. Additionally, the variety of healthcare professional types included and the diversity of the assessment tools used to measure healthcare professionals’ knowledge, attitudes, and practices may have affected the comparability of results across the HCPs. Finally, the lack of complete information on the reasons for non-participation by healthcare professionals may have introduced a selection bias.

Additionally, the predominance of US-based studies limits the applicability of findings to other healthcare systems. While many included studies offer valuable insights, cultural norms, clinical training structures, and models of adolescent care differ not only between high-income and low- and middle-income countries but also among high-income countries. For example, the organisation of primary care, school-linked services, and provider roles in adolescent health can vary between the US, Australia, the UK, and other European nations. Moreover, the US context is unique in its relatively high adolescent vaping prevalence and rapidly shifting regulatory environment. In 2024, approximately 1.63 million middle and high school students in the US reported current e-cigarette use, with 87.6% using flavoured products [41]. The US Food and Drug Administration (FDA) has implemented strict measures, including denying over a million marketing applications for flavoured e-cigarette products and enforcing a federal minimum legal sales age of 21 for all tobacco products [42]. These contextual factors influence both the attitudes and practices of US-based healthcare providers and should be considered when interpreting the findings. Therefore, as the majority findings from US studies, these systemic differences should be taken into account when applying these findings in different global contexts.

### 4.2. Future Directions

Future research should address the gaps identified in this review by exploring HCPs’ practices in diverse cultural and socioeconomic contexts in different countries. Additionally, research studies should design and evaluate targeted training programs and vaping-specific clinical guidelines that will provide HCPs with the skills to manage adolescents regarding vaping cessation support. HCPs’ practices over time can be observed by longitudinal and qualitative studies, and with the involvement of adolescents, we can capture both HCPs’ and adolescents’ perspectives to develop patient-centred interventions.

## 5. Conclusions

This review highlights the inconsistent knowledge and varied perceptions of HCPs regarding the safety of adolescent vaping. It also found that screening for vaping is not a routine practice. Given the increased prevalence of vaping amongst adolescents, there is an urgent need to integrate screening and counselling for vaping within routine clinical care. To address this, we recommend implementing standardised screening tools for vaping in adolescent health settings, adding dedicated vaping questions to electronic health records to prompt routine assessment, and developing communication aids tailored to adolescents that support risk education and behaviour change during clinical encounters. In addition, integrating vaping cessation training into medical and nursing education and ensuring its availability through accredited continuing professional development programs will better equip healthcare providers to address this growing public health issue. Findings from this review can guide the development of clinical guidelines, support tobacco control policy, and inform targeted adolescent health strategies aimed at reducing vaping-related harm.

## Figures and Tables

**Figure 1 ijerph-22-00839-f001:**
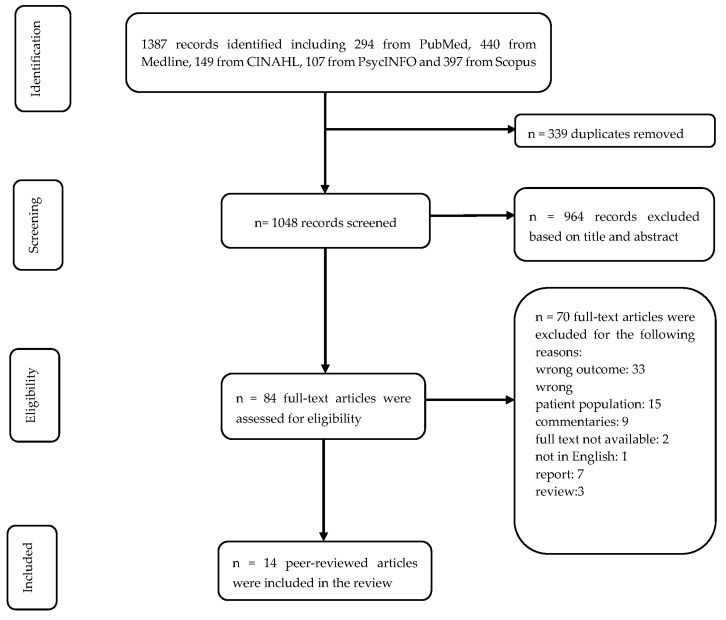
PRISMA flow chart of the study selection process.

**Table 1 ijerph-22-00839-t001:** Inclusion and exclusion criteria of the scoping review (PCC).

	Criterion	Inclusion	Exclusion
Population	Sample	HCPs involved in providing vaping cessation support to adolescents (e.g., medical doctors, general practitioners, paediatricians, nurses, dentists, pharmacists, psychologists, and allied health professionals)	Healthcare students, administrative staff, technicians, or professionals not directly engaged in vaping cessation support
	Geographical location	Not geographically restricted	N/A
	Setting	Any healthcare or community setting where HCPs deliver vaping cessation support (e.g., hospitals, primary care clinics, community health centers, schools)	Studies focused exclusively on settings serving adult populations or those not directly related to adolescent vaping cessation support
Concept	Study focus	Studies examining the knowledge, attitudes, and practices of HCPs regarding vaping cessation support for adolescents	Studies focusing on the use of e-cigarettes solely as a smoking cessation aid for adults, or those examining the personal vaping or smoking habits of HCPs
	Time period	Original peer-reviewed research published in English with no date restrictions	Not applicable
Context	Type of article	Original research articles published in peer-reviewed academic journals or scholarly sources	Commentaries, editorials, conference abstracts, letters to editors, and reviews
	Language	Studies published in English	Studies published in languages other than English

## Data Availability

Not applicable.

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
