# Peer review of "What Strategies Do Healthcare Providers Use to Promote Adolescents’ Vaping Cessation? A Scoping Review"

_ijerph, 2025, doi:10.3390/ijerph22060839_

Round 1

Reviewer 1 Report

Comments and Suggestions for Authors

I would like to thank the editor and the authors for giving me the opportunity to review this work. The article addresses a highly relevant and timely topic, particularly regarding the role of healthcare professionals in supporting adolescents in vaping cessation. The analysis conducted is well-structured and provides a clear overview of the practices and challenges that healthcare professionals face in dealing with the growing issue of adolescent vaping. The work highlights several important points that deserve attention, but there are also some areas that could be improved or clarified.

Strengths of the work:

  1. The method used, a scoping review based on the Arksey and O'Malley framework, is appropriate for a topic of this scope and allows for a comprehensive mapping of the main gaps in healthcare professionals' practices regarding adolescent vaping. The study selection, based on five major databases, and the number of articles included (14) are appropriate and well-justified, as stated in line 163.

  2. A particularly strong aspect is the detailed discussion of the barriers that healthcare professionals encounter, such as lack of knowledge, difficulties in screening, and limited time to properly address the issue of vaping. Lines 318-319 are particularly insightful regarding these challenges. Furthermore, the analysis of the beliefs of healthcare professionals across different specialties is crucial and adds value to the overall understanding of the issue, as highlighted in lines 334-336.

  3. The section on healthcare professionals' practices (lines 213-227) clearly highlights how screening and cessation support for vaping are neglected compared to traditional smoking. This is an important point for guiding future health policies.

  4. The need for training among healthcare professionals is clearly identified in several sections, with a particular focus on how training programs can significantly improve screening and counseling skills. Lines 237-240 show how training can significantly enhance healthcare professionals' comfort levels with screening and counseling adolescents for vaping.

Suggestions for improvement:

  1. Line 50: The phrase "Manufacturers continue to flourish the latest devices and attractive packaging on vaping products to target teenagers and young adolescents" could be more precise. I suggest rephrasing it as: "Manufacturers continue to promote new devices and attractive packaging for vaping products, which often target teenagers and young adolescents."

  2. Lines 98-99: "Systematic search strategy was developed in consultation with the university librarian to identify peer-reviewed articles" could benefit from more detail on the search process. For example: "A systematic search strategy was developed in consultation with the university librarian, using a comprehensive set of keywords to identify peer-reviewed articles across multiple databases."

  1. Although limitations are not explicitly mentioned, there are some implicit limitations that could be helpful to highlight. I would recommend adding a limitations section that could cover the following points:

    • Geographical: Most of the studies included come from the United States, and the lack of data from low- and middle-income countries could limit the generalizability of the results. It would be useful to comment on this limitation.
    • Variety in types of healthcare professionals (HCPs): Including a wide range of healthcare professionals makes it difficult to directly compare their practices. It would be helpful to reflect on how this diversity impacts the findings.
    • Heterogeneity of measurement tools: The tools used to assess knowledge, attitudes, and practices vary across studies, which makes it challenging to synthesize the results uniformly. Greater alignment in measurement tools would help improve the comparability of the data.

These limitations could be expressed in the manuscript as follows: "The main limitations of this study include the predominance of studies from the United States, with a lack of data from low- and middle-income countries. Additionally, the variety of healthcare professional types included and the diversity of the assessment tools used to measure healthcare professionals' knowledge, attitudes, and practices may have affected the comparability of results. Finally, the lack of complete information on the reasons for non-participation by healthcare professionals may have introduced a selection bias."

I would like to suggest that the authors include the following article, which could provide additional context and complement the discussion on adolescent behaviors and health interventions. The citation is as follows:

  • Diotaiuti P, Mancone S, Corrado S, De Risio A, Cavicchiolo E, Girelli L, and Chirico A (2022) Internet addiction in young adults: The role of impulsivity and codependency. Front. Psychiatry 13:893861. doi: 10.3389/fpsyt.2022.893861.

This citation could be included in line 72 as follows: "Prior studies by Belon et al. (2025) on prevention strategies, Wu (2024) on health messaging, and DiCasmirro (2024) on public health policies did not assess how HCPs counsel and intervene in adolescent vaping.

Diotaiuti et al. (2022) provided insights into the role of impulsivity and codependency in adolescent behaviors that may complement the findings of this review."

Comments on the Quality of English Language

The quality of the English language in the manuscript is generally strong, with clear and coherent expression throughout. However, there are some areas where minor adjustments could improve readability and precision. For example, certain sentences could be rephrased for better clarity, and there are a few instances where word choices or phrasing could be more concise.

Some technical terms could be defined or explained more explicitly for broader accessibility. Overall, the language is professional and suitable for the academic context, but slight revisions would enhance its overall flow.

Author Response

Comment 1. Line 50: The phrase "Manufacturers continue to flourish the latest devices and attractive packaging on vaping products to target teenagers and young adolescents" could be more precise. I suggest rephrasing it as: "Manufacturers continue to promote new devices and attractive packaging for vaping products, which often target teenagers and young adolescents."

Response1: Thank you for your suggestion, we have rephrased that line, the updated text now reads (Page: 2, Line: 69,70) "Manufacturers continue to promote new devices and attractive packaging for vaping products, which often target teenagers and young adolescents."

Comment 2: Lines 98-99: "Systematic search strategy was developed in consultation with the university librarian to identify peer-reviewed articles" could benefit from more detail on the search process. For example: "A systematic search strategy was developed in consultation with the university librarian, using a comprehensive set of keywords to identify peer-reviewed articles across multiple databases."

Response 2: Thank you for your insightful feedback. We have incorporated your suggestion and expanded the description of the search strategy. The revised sentence now reads: "A systematic search strategy was developed in consultation with the university librarian, using a comprehensive set of keywords to identify peer-reviewed articles across multiple databases." (Page: 3, Line:139-141)

Comment 3: Although limitations are not explicitly mentioned, there are some implicit limitations that could be helpful to highlight. I would recommend adding a limitations section that could cover the following points:

  • Geographical: Most of the studies included come from the United States, and the lack of data from low- and middle-income countries could limit the generalizability of the results. It would be useful to comment on this limitation.
  • Variety in types of healthcare professionals (HCPs): Including a wide range of healthcare professionals makes it difficult to directly compare their practices. It would be helpful to reflect on how this diversity impacts the findings.
  • Heterogeneity of measurement tools: The tools used to assess knowledge, attitudes, and practices vary across studies, which makes it challenging to synthesize the results uniformly. Greater alignment in measurement tools would help improve the comparability of the data.

These limitations could be expressed in the manuscript as follows: "The main limitations of this study include the predominance of studies from the United States, with a lack of data from low- and middle-income countries. Additionally, the variety of healthcare professional types included and the diversity of the assessment tools used to measure healthcare professionals' knowledge, attitudes, and practices may have affected the comparability of results. Finally, the lack of complete information on the reasons for non-participation by healthcare professionals may have introduced a selection bias.

I would like to suggest that the authors include the following article, which could provide additional context and complement the discussion on adolescent behaviors and health interventions. The citation is as follows:

  • Diotaiuti P, Mancone S, Corrado S, De Risio A, Cavicchiolo E, Girelli L, and Chirico A (2022) Internet addiction in young adults: The role of impulsivity and codependency. Front. Psychiatry 13:893861. doi: 10.3389/fpsyt.2022.893861.

This citation could be included in line 72 as follows: "Prior studies by Belon et al. (2025) on prevention strategies, Wu (2024) on health messaging, and DiCasmirro (2024) on public health policies did not assess how HCPs counsel and intervene in adolescent vaping.

Diotaiuti et al. (2022) provided insights into the role of impulsivity and codependency in adolescent behaviors that may complement the findings of this review."

Response 3:Thank you for your thoughtful and constructive feedback. We appreciate your suggestions for strengthening the manuscript. In response, we have incorporated the wording you suggested in the limitation section: the updated texts now reads,

The main limitations of this study include the predominance of studies from the United States, with a lack of data from low- and middle-income countries. (Page: 14, Line: 502,503)

Additionally, the variety of healthcare professional types included, and the diversity of the assessment tools used to measure healthcare professionals' knowledge, attitudes, and practices may have affected the comparability of results across the HCPs. (Page: 14, Line: 513-516)

Finally, the lack of complete information on the reasons for non-participation by healthcare professionals may have introduced a selection bias. (Page: 14, Line: 516-518)

  • Regarding the article: Thank you for this thoughtful suggestion. We appreciate the relevance of Diotaiuti et al. (2022) in highlighting psychological factors such as impulsivity and codependency in adolescent behaviours. However, after careful consideration, we have decided not to include this citation in the current version of the manuscript, as the primary focus of our review remains on healthcare providers’ roles, clinical practices, and systemic barriers in vaping cessation support. While psychological aspects are indeed important, they are beyond the specific scope of this review. However, we acknowledge the value of this perspective and will consider integrating such behavioural dimensions in future work or subsequent publications.

Comments on the Quality of English Language

The quality of the English language in the manuscript is generally strong, with clear and coherent expression throughout. However, there are some areas where minor adjustments could improve readability and precision. For example, certain sentences could be rephrased for better clarity, and there are a few instances where word choices or phrasing could be more concise.

Some technical terms could be defined or explained more explicitly for broader accessibility. Overall, the language is professional and suitable for the academic context, but slight revisions would enhance its overall flow.

Response: Thank you for your positive feedback on the overall quality of the manuscript. We have carefully reviewed the manuscript in response to your comments to improve clarity, precision, and readability. Minor rewording has been applied to several sentences for better flow, and technical terms have been more clearly defined to ensure accessibility to a broader audience. We believe these revisions have further strengthened the manuscript

Reviewer 2 Report

Comments and Suggestions for Authors

This manuscript is a useful contribution to the literature on electronic cigarette use among youth and the need for cessation therapy for this population, with a focus on health care providers and how their perceptions of youth vaping may impede cessation.  The authors are careful to recount the various study limitations of their scoping review, and they also describe their methods clearly.  No major critiques from this reviewer.

Minor, easily fixable points: the authors place some reference numbers before the period and others after; they should be consistent.  See for example page 2, line 75-79.  Oddly, simultaneously, they use author date references even as they use numbered references.  Again, these should be made uniform.  Throughout, there are some periods with no space after them; the next sentence is jammed up against them.

  1. 2 line 66: there should be no comma after HCPs.
  2. 2 line 92 there should be a d in aolescents.
  3. 5 line 192. The sentence starting with Whilst is an incomplete sentence. Therefore, it should be preceded by a comma instead of a period.

Author Response

Comment1:

Minor, easily fixable points: the authors place some reference numbers before the period and others after; they should be consistent.  See for example page 2, line 75-79.  Oddly, simultaneously, they use author date references even as they use numbered references.  Again, these should be made uniform.  Throughout, there are some periods with no space after them; the next sentence is jammed up against them.

Reponse 1:

Thank you for highlighting these important formatting issues. We have revised the manuscript throughout to maintain a consistent numbered citation style. Reference numbers are now placed before the period, and instances of mixed citation styles have been corrected. (Page: 3, Line: 109-112)

Minor typographical issues have also been addressed, such as missing spaces after periods

Comment 2:

  1. 2 line 66: there should be no comma after HCPs.
  2. 2 line 92 there should be a d in aolescents.
  3. 5 line 192. The sentence starting with Whilst is an incomplete sentence. Therefore, it should be preceded by a comma instead of a period.

Response 2:

Thank you for identifying these minor errors. We have carefully addressed each point as follows:

1.The unnecessary comma after "HCPs" on line 66 has been removed. (Page:2 , Line: 82)

2.The spelling error in "aolescents" has been corrected to "adolescents" on line 92. (Page: 3, Line:132)

3.The punctuation issue on line 192 has been corrected by changing the period to a comma before "Whilst," ensuring proper sentence structure. (Page: 6, Line: 267)

Reviewer 3 Report

Comments and Suggestions for Authors

The topic of the manuscript is of interest, as it is absolutely true that more adolescents and adults are vaping, and there is no clear trend toward its prevention in the countries with the highest vaping rates.
The review is very understandable, and the information regarding the methodology and results is clear. The information included in the tables is understandable and complements the results appropriately. The bibliography is up-to-date.

The aims of the study titled: “What strategies do healthcare providers use to support adolescents for vaping cessation? A scoping review” is to assess knowledge, attitude,
and practices of HCPs regarding vaping cessation support to adolescents. The authors have followed all the steps recommended from the perspective of the Arksey and O'Malley framework explained below:

 The study identifies the research question and consider how the question
informs the search strategy: Specify population, concept, and context: What strategies do healthcare providers use to support adolescents for vaping cessation?
 There are relevant literature and include published articles related to the
main topic of the review. Moreover, the authors define the inclusion and exclusion criteria after understanding of the scope of the literature. In addition, using independent reviewers to apply inclusion and exclusion criteria. Include the Preferred Reporting Items for Systematic Reviews 
and Meta-Analyses (PRISMA) flow Chart. Specifically, one independent reviewer (NN) conducted the titles and abstracts screening. Three reviewers (NN, AR, and BJ) independently conducted the full-text review of the eligible articles. We can see the PRISMA flow chart of the 
study selection process (Figure 1).
 The authors extract, mapping, and charting the data and include quantitative and qualitative analysis: The authors included original peer-reviewed articles published in English, irrespective of their design (quantitative or qualitative). The first author (NN) carried out the data 
extraction based on the research question and reviewed by other co-authors. The quality of the selected papers was assessed using the Critical Appraisal Skills Program.
 The authors summarize, synthesize, and report the results: In this research, three authors independently reviewed all full-text articles, including them in the final selection based on predefined criteria. The authors point out that any disagreements or doubts were resolved 
through discussion amongst the co-authors. This review included ten cross-sectional, two qualitative, one mixed-method, and one retrospective cohort study. We can see a Summary characteristics of the included studies (Table 2) and Supplementary Materials. In my opinion
there are a very complete discussion, with specific sections: Strength, Limitations and Future direction

Comments: I have detected some typographical errors, joined words, extra spaces between words, etc., both in the text and in the supplementary Table 1. (e.g., line 92 in the text).

Author Response

Comment :

I have detected some typographical errors, joined words, extra spaces between words, etc., both in the text and in the supplementary Table 1 (e.g., line 92 in the text).

Response:

Thank you for your careful review. We have carefully reviewed and corrected all typographical errors and formatting inconsistencies in the main text and Supplementary Table 1.

We have corrected line 92, revised line in (Page: 3, Line: 132)

Reviewer 4 Report

Comments and Suggestions for Authors

Nimmi and coworkers searched over a thousand primary literature articles related to HCP advising adolescent patients about vaping. They only included 14 of these studies in their manuscript. Authors mention in Figure 1 that some studies were excluded based on "wrong outcome".  Please include one more statement in the materials and methods where you explain this point in more detail. Language is another unfortunate and limiting reason why other studies would be excluded. In the discussion, authors mention that the 14 studies are all from US. Nonetheless, authors do state this in the "limitations" section. 

There are a few typos and spacing issues here and there, that I am sure the assistant editor will fix as the manuscript is processed.

In my opinion, this review paper makes an excellent point of the major obstacles that HCP face when advising about vaping. It is extremely important that HCPs are better informed on the primary lit in this field, and I agree this would be a monumental task to put this responsibility on HCPs only. I hope when this review paper is published, it will encourage healthcare administrators to educate HCPs.

I would like to end by thanking the authors for summarizing the current knowledge in this topic and emphasizing the need for professional guidelines to support HCPs.

Author Response

Comment 1:

Nimmi and coworkers searched over a thousand primary literature articles related to HCP advising adolescent patients about vaping. They only included 14 of these studies in their manuscript. Authors mention in Figure 1 that some studies were excluded based on "wrong outcome".  Please include one more statement in the materials and methods where you explain this point in more detail. Language is another unfortunate and limiting reason why other studies would be excluded. In the discussion, authors mention that the 14 studies are all from US. Nonetheless, authors do state this in the "limitations" section. 

Response 1:

Thank you very much for your constructive and encouraging feedback. In response to your suggestion, we have added a clarification in the Materials and Methods section under "Inclusion and Exclusion Criteria." Specifically, we now explain that studies were excluded under "wrong outcome" if they did not focus on healthcare providers' views, practices, or support strategies for adolescent vaping cessation. Studies were also excluded if they focused only on adult smoking cessation, the smoking behaviours of HCPs themselves, or provided outcomes that were irrelevant, incomplete, or incompatible with the aim of the review. (Page: 4, Line: 192-196)

Moreover, we have also included in the limitation section that reads “The main limitations of this study include the predominance of studies from the United States, with a lack of data from low- and middle-income countries.” (Page: 14, Line: 502-503)

Comment 2:

There are a few typos and spacing issues here and there, that I am sure the assistant editor will fix as the manuscript is processed

Response 2:

Thank you for your observation. We appreciate your attention to detail. We have carefully reviewed the manuscript once again and made necessary corrections to address any typographical or spacing inconsistencies. We are also confident that the editorial team will ensure the final formatting aligns with the journal’s standards during the manuscript process. Thank you for your understanding and support.

Reviewer 5 Report

Comments and Suggestions for Authors

Dear Authors,

Thank you for the opportunity to review your manuscript. While your study addresses an important topic in public health, there are several substantial areas that require critical revision to improve the clarity, depth, and scholarly value of the paper. The comments below aim to give helpful suggestions to strengthen the manuscript.

The title, while descriptive, could benefit from greater precision. The term “support” is broad and unspecific. Consider narrowing or qualifying it to reflect whether the focus is on screening, counselling, cessation advice, or all three.

The abstract lacks specificity in summarizing the study’s findings. It currently provides general statements about variation in knowledge and attitudes but fails to offer measurable patterns or key insights. It would benefit from the inclusion of quantitative indicators (e.g., how many studies reported inadequate screening or specific gaps in provider confidence).

The introduction lacks conceptual framing. While it discusses the rise of vaping, it does not provide a theoretical rationale for the role of HCPs nor critically synthesize the existing gaps in knowledge.

Several references (e.g., those on policy or global prevalence) are used in a narrative fashion without clearly linking them to the core research problem or the role of healthcare professionals.

The search strategy is not fully reproducible. Although databases and keywords are listed, search strings are not detailed, and Boolean logic is not presented. PRISMA-ScR adherence should include transparent reporting of the full search syntax.

The selection process lacks clarity in operational terms. While inclusion and exclusion criteria are presented, the way terms like “support” or “adolescents” were defined across studies is not clear.

The critical appraisal process using CASP is only superficially described. The method for synthesizing appraisal ratings (e.g., A/B/C grades) is unclear, and the decision-making process for resolving disagreements among reviewers is vaguely stated.

Consultation with stakeholders, a key element of the Arksey and O’Malley framework, is omitted with limited justification. Even if not conducted, the rationale should be strengthened and the implications of excluding this step discussed.

The results section lacks analytical depth. It reads as a narrative summary without a structured synthesis. There is little effort to stratify findings by type of healthcare provider, geographic location, or methodology of included studies.

The reporting of findings is descriptive rather than interpretive. The manuscript frequently states that knowledge was “mixed” or that providers were “concerned” without offering comparative or pattern-based interpretations across the sample.

Heterogeneity in measurement tools and outcome definitions is not adequately addressed. This omission weakens the reader’s ability to critically assess the reliability of cross-study comparisons.

The discussion section is too general and fails to integrate findings into a conceptual framework. It restates results rather than analyzing their implications in the context of existing research or clinical practice.

Training gaps are acknowledged but underdeveloped. The discussion does not explore why current training fails or which competencies are most lacking. A critical discussion of the structural barriers to implementing training in healthcare systems is missing.

The limitations section does not sufficiently address methodological shortcomings. For instance, the reliance on predominantly U.S.-based studies is mentioned, but there is no discussion of how cultural or system-level differences may influence the applicability of findings.

The policy relevance of the findings is not clearly articulated. The review misses the opportunity to discuss how these findings could inform clinical guidelines, public health campaigns, or adolescent health policy.

The conclusion repeats general ideas but does not provide clear, practical recommendations. It would be stronger if it included specific strategies like using standard screening tools, adding vaping questions to health records, and creating communication tools for adolescents.

Author Response

Comment 1:

The title, while descriptive, could benefit from greater precision. The term “support” is broad and unspecific. Consider narrowing or qualifying it to reflect whether the focus is on screening, counselling, cessation advice, or all three.

Response 1:

Thank you for this constructive suggestion. In response, we have revised the title to enhance specificity and better reflect the scope of the review. The updated title, “What strategies do healthcare providers use to promote adolescent vaping cessation? A scoping review”, clarifies the focus on strategies aimed at vaping cessation and aligns more closely with the core themes explored in the study, including screening, counselling, and clinical interventions. We appreciate your recommendation, which has helped improve the clarity and precision of the manuscript. (Page: 1, Line: 2- 3) .

Comment 2:

The abstract lacks specificity in summarizing the study’s findings. It currently provides general statements about variation in knowledge and attitudes but fails to offer measurable patterns or key insights. It would benefit from the inclusion of quantitative indicators (e.g., how many studies reported inadequate screening or specific gaps in provider confidence).

Response 2:

Thank you. We have revised the abstract to include key quantitative indicators, such as the proportion of HCPs who screened for vaping (14% vs. 86% for cigarettes) and the range of knowledge gaps (reported between 50%-92% across studies). (Page: 1, Line: 26-29)

Comment 3:

The introduction lacks conceptual framing. While it discusses the rise of vaping, it does not provide a theoretical rationale for the role of HCPs nor critically synthesize the existing gaps in knowledge

Response 3:

Thank you for this valuable comment. We acknowledge the importance of conceptual framing in clarifying the relevance of this review. The scope of this review did not include the application of psychological theories, as the included studies did not report the use of such frameworks. However, we recognise that behavioural models such as the Theory of Planned Behaviour (TPB) and Social Cognitive Theory (SCT) may offer useful lenses to explain variations in HCPs engagement in vaping cessation efforts. Specifically, these theories suggest that attitudes, perceived norms, confidence, and self-efficacy may influence whether HCPs act on opportunities for intervention. We now briefly acknowledge this in the Introduction while noting that future empirical research is needed to explore these theoretical explanations in greater detail. (Page: 2 and 3, Line: 92-99).

We have also strengthened the Introduction by better positioning HCPs as key actors in preventive adolescent care and synthesising key gaps in current knowledge, including limited clinical guidance and training. These additions enhance the theoritical rationale for undertaking this review. (Page: 2 and 3, Line: 87-92,100-105).

Comment 4:

Several references (e.g., those on policy or global prevalence) are used in a narrative fashion without clearly linking them to the core research problem or the role of healthcare professionals.

Response 4:

Thank you for this helpful comment. We have revised the Discussion to provide clearer interpretation of findings within clinical, educational, and policy contexts. The revised section now explains why current training often fails due to reliance on adult-focused models, lack of adolescent-specific competencies, and limited integration into professional development. We also highlight structural barriers such as time constraints and the absence of national guidelines. To strengthen policy relevance, we recommend developing clinical guidelines, standardised screening tools, vaping-related prompts in health records, and aligned public health messaging across clinical and school-based settings. (Page: 13, Line: 469-492)

We want to clarify how global references relate to the core research problem. We included international prevalence data to provide context for the scope of adolescent vaping as a global public health concern. These statistics emphasise that vaping issue is not limited to a specific region, highlighting the urgency for healthcare systems, including those in Australia, to respond effectively.

Comment 5:

The search strategy is not fully reproducible. Although databases and keywords are listed, search strings are not detailed, and Boolean logic is not presented. PRISMA-ScR adherence should include transparent reporting of the full search syntax.

Response 5:

Thank you. We have provided the full search strings, including Boolean operators, in Supplementary Table S3, and referenced this addition in the Methods section. (Page: 4, Line: 172)

Comment 6:

The selection process lacks clarity in operational terms. While inclusion and exclusion criteria are presented, the way terms like “support” or “adolescents” were defined across studies is not clear

Response 6:

Thank you for the suggestion. We have revised the Inclusion and Exclusion Criteria section to clearly define the terms “support” and “adolescents” used in our review. These definitions ensure consistent interpretation and enhance the clarity of our selection process. (Page: 4, Line: 180-185)

Comment 7:

The critical appraisal process using CASP is only superficially described. The method for synthesizing appraisal ratings (e.g., A/B/C grades) is unclear, and the decision-making process for resolving disagreements among reviewers is vaguely stated.

Response 7:

Thank you for your helpful comment. We have expanded the Quality Appraisal section to clearly describe the application of the CASP tool, including how overall study ratings (A/B/C/D) were assigned, and how discrepancies between reviewers were resolved through consensus discussion. (Page: 5, Line: 219-229)

Comment 8:

Consultation with stakeholders, a key element of the Arksey and O’Malley framework, is omitted with limited justification. Even if not conducted, the rationale should be strengthened and the implications of excluding this step discussed.

Response 8:

Thank you for this valuable comment. Stakeholder consultation, while an important optional step in the Arksey and O’Malley framework, was not conducted as the primary aim was to map peer-reviewed literature rather than develop practice recommendations. We have strengthened the justification in the Methods section and acknowledged the exclusion of stakeholder consultation as a limitation that may affect broader contextual insights. While we did not undertake a formal stakeholder consultation process as part of this scoping review, our investigator team included a paediatrician and a general practitioner who were actively involved throughout the development, analysis, and interpretation stages. Their clinical expertise and practical insights provided meaningful input that helped contextualise the review. (Page: 4, Line: 149-159) 

Comment 9:

The results section lacks analytical depth. It reads as a narrative summary without a structured synthesis. There is little effort to stratify findings by type of healthcare provider, geographic location, or methodology of included studies.

Response 9:

Thank you. We have restructured the Results section with subheadings to stratify findings by healthcare provider type and geographic location. (Page: 6, Line: 278-284), (Page: 7, Line: 323-328),

Comment 10:

The reporting of findings is descriptive rather than interpretive. The manuscript frequently states that knowledge was “mixed” or that providers were “concerned” without offering comparative or pattern-based interpretations across the sample.

Response 10:

Thank you for this observation. We have revised the Results section to provide a more structured synthesis and interpretation of findings, moving beyond general descriptions. Specifically:

  • We stratified findings by provider type (e.g., GPs, paediatricians, nurse practitioners) to highlight variations
  • We compared geographic trends, with studies from the USA and Australia separately to show how location influenced clinical behaviours.
  • We interpreted the influence of training exposure, noting that trained providers demonstrated higher screening and counselling rates than their untrained counterparts.
  • We replaced generic phrases such as “mixed knowledge” or “general concern” with specific statistical ranges (80% awareness of nicotine content) and termed them as beliefs rather than concerns.
  • These changes are reflected in revised subsections 3.2.1. (Page: 6, Line: 278-284), 3.4.1 (Page: 7, Line: 323-328) and in 3.5 respectively of the Results and are further integrated into the Discussion section to enhance interpretive depth.

Comment 11:

Heterogeneity in measurement tools and outcome definitions is not adequately addressed. This omission weakens the reader’s ability to critically assess the reliability of cross-study comparisons.

Response 11:

Thank you for this helpful comment. We agree that heterogeneity in measurement tools and outcome definitions limited the comparability of findings across studies. To address this:

  • We have clarified in the Results (Section 3.1) that the included studies used a wide range of measurement tools—from Likert scales to open-ended interviews—with varying outcome definitions. Knowledge, counselling, clinical skills and inconsistent age group classifications. (Page: 5, Line: 242-253)
  • Under the Discussion section in the Strength and Limitations subsection, we now explicitly acknowledge this heterogeneity as a methodological limitation that reduces the ability to perform direct cross-study comparisons or draw generalisable conclusions. (Page: 14, Line: 502-511).
  • We also note that this variation underscores the need for standardised assessment tools in future research on HCP support for vaping cessation. (Page: 14, Line: 511-518).
  • These additions aim to improve transparency and help readers better interpret the consistency and reliability of the findings.

Comment 12:

The discussion section is too general and fails to integrate findings into a conceptual framework. It restates results rather than analyzing their implications in the context of existing research or clinical practice.

Response 12:

Thank you for this valuable comment. In response, we have revised the Discussion section to go beyond restating the findings and provide a clearer interpretation in the context of existing literature and clinical practice. While this review did not apply psychological theories in depth consistent with the scope and nature of a scoping review, we now briefly relate our findings to key behavioural constructs, such as attitudes, norms, and self-efficacy, which are reflected in the Theory of Planned Behaviour (TPB) and Social Cognitive Theory (SCT). These theoretical perspectives may help explain why some HCPs engage in adolescent vaping cessation while others do not, despite similar clinical exposure. We acknowledge that future empirical research is needed to examine these models more rigorously. (Page: 12 and 13, Line: 443-451)

We have also expanded our interpretation of how limited training, unclear clinical pathways, and lack of guidelines affect healthcare providers' preparedness and engagement. These changes strengthen the relevance of our findings to current practice and policy discussions. (See revised Discussion, Page 13, Line 457-492)

Comment 13:

Training gaps are acknowledged but underdeveloped. The discussion does not explore why current training fails or which competencies are most lacking. A critical discussion of the structural barriers to implementing training in healthcare systems is missing.

Response 13:

Thank you for this insightful comment. We have revised the Discussion section to more critically examine the limitations of current training approaches and identify key competency gaps specifically: (See revised Discussion, Page 13, Line 457-492)

  • We now discuss how existing training often lacks adolescent-specific content and is limited in scope (e.g., focused on tobacco rather than vaping), which restricts its relevance and applicability to current clinical challenges.
  • We identify specific competencies that are most lacking, including risk communication with adolescents, screening and counselling skills for vaping, and knowledge of referral pathways or pharmacotherapy options.
  • We also expand on the structural barriers to implementing training, including time constraints in clinical settings and continuing professional development.

Comment 14:

The limitations section does not sufficiently address methodological shortcomings. For instance, the reliance on predominantly U.S.-based studies is mentioned, but there is no discussion of how cultural or system-level differences may influence the applicability of findings.

Response 14:

Thank you for this important observation. The predominance of U.S.-based studies may limit the generalisability of our findings. In response, we have expanded the Strengths and Limitations section of the Discussion to provide a more transparent appraisal of this issue. Specifically, we now discuss how cultural norms, healthcare financing structures, models of adolescent care, and public health infrastructure differ not only between high-incomeand low- and middle-income countries but also among high-income countries themselves.

We also provide further context on the current status of adolescent vaping and the regulatory landscape in the United States, including recent prevalence estimates and FDA policy actions. This contextualisation helps clarify why U.S.-based findings must be interpreted cautiously when considering applicability to other settings. (Page 14, Line 519-536)

Comment 15:

The policy relevance of the findings is not clearly articulated. The review misses the opportunity to discuss how these findings could inform clinical guidelines, public health campaigns, or adolescent health policy.

Response 15:

Following your insightful suggestion, we have revised the Discussion and Conclusion sections to more clearly articulate the policy implications of our findings. (See revised Discussion, Page 13, Line 457-492) (Page 15, Line 567-576)Specifically:

  • We now highlight the potential for these findings to inform the development of national clinical guidelines for adolescent vaping cessation, mainly through structured screening tools, referral pathways, and provider training recommendations.
  • We also discussed how the gaps identified in provider knowledge and confidence can inform the design of public health campaigns that align messaging across clinical and community settings.
  • We emphasise the importance of integrating vaping cessation into existing adolescent health policy frameworks, including preventive health checkups, school-based health programs, and workforce development strategies.

Comment 16:

The conclusion repeats general ideas but does not provide clear, practical recommendations. It would be stronger if it included specific strategies like using standard screening tools, adding vaping questions to health records, and creating communication tools for adolescents.

Response 16:

Thank you for this constructive comment. We have revised the Conclusion section to include specific, actionable strategies informed by our findings. These include the need for standardised screening tools to be adopted in adolescent health settings, the integration of vaping-specific questions into electronic health records to prompt routine assessment, and the development of communication resources tailored to adolescent patients to support brief interventions in clinical practice. (Page 15, Line 567-576)

Round 2

Reviewer 1 Report

Comments and Suggestions for Authors

"After a careful review and the implementation of the suggested revisions, the article complies with the editorial requirements and is now ready for publication."

Reviewer 5 Report

Comments and Suggestions for Authors

Thank you for your thorough and thoughtful revisions. I appreciate the effort you have put into addressing the comments and suggestions provided in the initial review. Your responses are clear and well-justified, and the manuscript has improved in terms of clarity, methodology, and overall presentation. I have no further concerns or comments at this stage.

Wishing you success with this publication.